# Inline Reticulorumen pH as an Indicator of Cows Reproduction and Health Status

**DOI:** 10.3390/s20041022

**Published:** 2020-02-14

**Authors:** Ramūnas Antanaitis, Vida Juozaitienė, Dovilė Malašauskienė, Mindaugas Televičius

**Affiliations:** 1Large Animal Clinic, Veterinary Academy, Lithuanian University of Health Sciences, Tilžės str 18, Kaunas LT44307, Lithuania; dovile.malasauskiene@lsmuni.lt (D.M.); mindaugas.televicius@lsmuni.lt (M.T.); 2Department of Animal Breeding, Veterinary Academy, Lithuanian University of Health Sciences, Tilžės str 18, Kaunas LT44307, Lithuania; vida.juozaitiene@lsmuni.lt

**Keywords:** blood gas, reticulorumen, precision livestock farming (PLF), automatic milking system (AMS)

## Abstract

Our study hypothesis is that the interline registered pH of the cow reticulum can be used as an indicator of health and reproductive status. The main objective of this study was to examine the relationship of pH, using the indicators of the automatic milking system (AMS), with some parameters of cow blood components. The following four main groups were used to classify cow health status: 15–30 d postpartum, 1–34 d after insemination, 35 d after insemination (not pregnant), and 35 d (pregnant). Using the reticulum pH assay, the animals were categorized as pH < 6.22 (5.3% of cows), pH 6.22–6.42 (42.1% of cows), pH 2.6–6.62 (21.1% of cows), and pH > 6.62 (10.5% of cows). Using milking robots, milk yield, fat protein, lactose level, somatic cell count, and electron conductivity were registered. Other parameters assessed included the temperature and pH of the contents of reticulorumens. Assessment of the aforementioned parameters was done using specific smaX-tec boluses. Blood gas parameters were assessed using a blood gas analyzer (EPOC (Siemens Healthcare GmbH, Erlangen, Germany). The study findings indicated that pregnant cows have a higher pH during insemination than that of non-pregnant ones. It was also noted that cows with a low fat/protein ratio, lactose level, and high SCC had low reticulorumen pH. They also had the lowest blood pH. It was also noted that, with the increase of reticulorumen pH, there was an increased level of blood potassium, a high hematocrit, and low sodium and carbon dioxide saturation.

## 1. Introduction

The first widely adopted application of precision livestock farming (PLF), years before the term PLF was introduced, was the individual electronic milk meter [1]. The term PLF was coined in the early 1970s and 1980s. The other most commonly used parameters in PLF include the use of commercialized behavior based on estrus detection [2], rumination tags [3], and the use of an online milk time analyzer [4]. The sensors in these applications provide useful data that can be used by farmers to identify livestock that need special care before health conditions worsen [5]. One of the most accurate data sources used for continuous monitoring of individual livestock health status is the reticuloruminal pH (RRpH). The advantage of utilizing RRpH is due to its diurnal recording ability. Various scientific investigations have used continuous measurements of ruminal pH to assess livestock health status [6]. The technique entails the use of a memory chip inserted in the livestock’s rumen, and to retrieve the data, it has to be physically removed or an external cable is used to transmit data to an external unit.

According to Cantor [7], the use of reticulorumen temperature is an effective measure to predict livestock health status, such as via dairy herd water intake. Cantor argues that real-time observations of reticulorumen pH and temperature in fresh dairy cows are effective in assessing the risk of subclinical ruminal acidosis (SARA) because they provide an opportunity to evaluate the prophylactic effect of the treatment strategies applied [7]. Antanaitis [8] argues that some blood parameters and dairy cow rumination times can be used as indicators to accurately diagnose subclinical acidosis and ketosis. However, there is limited information on how the two parameters can be used to assess disease, so future studies should compare data findings using many animals. Over the last few decades, there has been a dramatic decrease in dairy cow fertility rate due to various preventable causes [9]. Reticuloruminal pH data can also be used to predict the reproductive health of livestock [10]. Dairy cows with altered rumen metabolism (that is, low pH) have low fertility rates. Therefore, using reticuloruminal pH is a great predictor of a dairy cow’s reproductive success. However, more studies on the role of reticuloruminal pH in determining cow reproductive health are needed [10]. Alzahal et al. assessed the ruminal temperature and pH of dairy cows and their association in predicting dairy cow nutritional and health status [11]. Similar studies conducted by Cooper-Prado et al. reported that ruminal temperature lowers one day prior to parturition [12]. Optimum diet fermentation and fiber digestion are achieved at a ruminal pH between 6.0 and 6.4. At this pH level, the cellulolytic bacteria effectively digest fiber, which is inhibited in pH levels below 6.0 [13]. Therefore, a decrease in ruminal pH increases acidity, which in turn increases the temperature of the abomasum [14]. Thus, by using the two parameters, one can predict the health status of a cow.

The two parameters/data are gathered using wireless sensor nodes that are often attached to the animal. The wireless sensors are then attached to wireless health monitoring systems. Analysis of the data collected can be used to assess, detect, and prevent numerous livestock diseases. Another method of collecting data is the use of rumen fluid samples, whereby the samples are collected using an oral–ruminal probe or rumen fistula. [15]. Rumen pH and temperature parameters fluctuate. However, the collection of rumen fluid samples should be avoided when possible because it causes distress to the research subjects [16]. With technological advancements, new noninvasive technologies, such as the use of intra-ruminal boluses, have been developed to collect pH and temperature data to monitor a cow intra-ruminal metabolism. However, there is limited information on how the interline registered reticulorumen pH can be utilized as an indicator to assess cow health status and reproductive systems. This study hypothesizes that interline registered reticulorumen pH can accurately predict cow reproduction and health status. The main objective of this study is to examine the relationship of reticulorumen pH with indicators and compare the automatic milking system (AMS) and blood indicators to determine the reproduction and health status of dairy cows.

## 2. Materials and Methods

### 2.1. Location and Experimental Design

The experiment was conducted on a dairy cow farm located in the Eastern part of Europe (54.9587408, 23.784146). About 95 Lithuanian black and white dairy cows that matched the selection criteria were identified. The inclusion criteria were cows that had two or more lactations. The cows needed to be identified as clinically healthy, have a temperature of 38.8 degrees Celsius, 5–6 rumen contractions every three minutes, and no signs and symptoms of laminitis, metritis, or mastitis. The research subjects were taken to an accommodation with loose-housing system where they were put on a constant feeding rotation during the entire research period. Nutritional balance was maintained to ensure that physiological needs were adequately met. The TMR comprised 30% corn silage, 4% hay grass, 50% grain mash concentrate, and 10% grass silage. This diet was formulated using NRC 2001 guidelines for a 550 kg Holstein cow producing 35 kg/d. The composition ration was as follows: DM (%)—48.8, NEL (Mcal/kg) 1.6; NDF, ADF, NFC, and CP percentage of DM was 28.2, 19.8, 38.7, and 15.8 respectively. Using this aforementioned mixed ration, the research subjects were fed twice a day at 10:00 h and 20:00 h. Two kilograms per day of concentrate was used to feed the cows at the milking site. The average BCS used was 3.45 (±0.25).

### 2.2. Measurements

SmaX-tec boluses (smaXtec animal care GmbH, Graz, Austria) were used to assess the content of cow reticulorumen pH and temperature. This device was preferred for this study because of its ability to display real-time pH and temperature data. Using the instruction manual, boluses were put into the cows’ reticulorumen. The data were collected using specific antennas on the SmaX-tec device. The boluses in the cows’ reticulorumens from 2–9 January 2019. Reticuloruminal pH was evaluated using an indwelling wireless data transmitting system (smaXtec). The entire system was controlled by a microprocessor. After conversion using an AD converter, the data was stored in an external memory chip. The device size was small enough to permit oral administration to an adult cow. More so, it was resistant to rumen fluid. At the beginning of the study, pH probes were calibrated using pH 4 and pH 7 buffers.

Lely Astronaut^®^ A3 milking robots were used to milk the cows. The robots were also used to register rumination time (RT) (min/d), yield MY (kg/d), bodyweight BW (kg), lactose ration (%), milk fat/protein ratio (F/P), milk electrical conductivity of all quarters of the udder (front left and right, EC1 and EC2, respectively; rear left (EC3) and right (EC4), respectively, in mS/cm), and conception of concentrates. Blood gas samples were obtained and stored in an ice bath until processing. Using Epoc blood gas analyzers (EPOC, Canada), the following parameters were obtained: base excess in blood (BE), partial carbon dioxide pressure (PCO2), partial oxygen pressure (PO2), bicarbonate (Chco3), Hydrogen potential (pH), total carbon dioxide carbon (TCO2), base excess in extracellular fluid (BE ecf), Sodium (Na), Calcium (C), Potassium (K), hematocrit (HCT), chlorides (cl), hemoglobin concentration (cHgb), and lactate (Lac).

### 2.3. Animals and Experimental Condition

The dairy cow reproductive system is classified as follows (Table 1):

According to the reticulorumen pH assay, the experimental animals were divided into four classes: 1. pH < 6.22 (5.3% of cows), 2. pH 6.22–6.42 (42.1% of cows), 3. pH 6.42–6.62 (21.1% of cows), and 4. pH > 6.62 (10.5% of cows). Estrus was identified with specific devices in this study measuring activity in steps, and rumination time (min/d) (by increasing activity and decreasing rumination time) was monitored by the herd management program, Lely Astronaut^®^ (24/7). The research subject was considered estrus according to the following parameters: restlessness, type and amount of mucous discharge, extent of alertness, tail raising, and congestion of the mucous membrane around the vulvar area. Uterine tone was assessed using rectal palpations. About 12 h after estrus signs were presented, the research subjects were inseminated using frozen semen. Successful implantation and pregnancies were confirmed using an Easi-Scan ultrasound device (IMV imaging, Scotland, UK) once around day 30 to 35. The pregnant cows were grouped in a different group from the non-pregnant ones.

### 2.4. Data Analysis and Statistics

Statistical data analysis was conducted using SPSS 20.0 (SPSS, Inc., Chicago, IL, USA) software. The data were then presented using descriptive statistics and normal distribution analysis methods, such as the Kolmogorov–Smirnov test. The statistical relationship between reticulorumen pH and AMS indicators, body weight (BW), activity of cows, milk yield (MY), milk fat/protein ratio (F/P), somatic cell count in milk (SCC), milk lactose content, and electrical conductivity of all four udder quarters were shown using Pearson correlations. To effectively analyze SCC variables, they were converted to SCClog 10. Analysis of the linear relationship between reticulorumen pH and the analyzed AMS was done using Pearson correlation. Multiple comparisons of groups means were calculated using Tukey’s test. A probability below 0.05 was considered reliable (*p*-Value < 0.05).

All the data were registered on the investigation day, except for pregnant and non-pregnant cows, whose data were registered on the insemination day.

## 3. Results

We determined that the average pH of the reticulorumen was 6.47 ± 0.016, temperature of the reticulorumen was 38.779 ± 0.020 °C, and rumination time was 455.26 ± 6.052. The average milk productivity of cows was 40.41 ± 0.724 kg, BW was 648.37 ± 13.265 kg, and the ratio of fat to protein in milk was 1.17 ± 0.013.

### 3.1. Reticulorumen pH as an Indicator of Reproduction Status of Cows

Analysis of the reticulorumen pH of cows by reproductive status showed the highest average value of this indicator in Group IV (Figure 1A), which was 2.13% higher compared to Group I, 0.76% higher compared to Group II, and 1.37% higher compared to Group III. According to multiple comparisons of means, all differences between the groups of cows by reproductive status were found to be statistically significant (*p* < 0.05).

We found (Figure 1B) that all pregnant cows (Group IV, n = 20) belonged to the third class according to their reticulorumen pH, which ranged between 6.42 to 6.62 (50.00% of the animals in this class were Group III cows).

The data in Figure 2A show that the pH of the first group (15–30 days postpartum) changed from 6 to 6.98 during the day. The range of changes in this indicator was 2–2.24 times higher compared to cows in the other groups.

On comparing the reticulorumen pH in non-pregnant and pregnant cows 35–90 days after insemination, we see a higher level of this indicator in pregnant cows.

### 3.2. Reticulorumen pH as an Indicator of Health Status in Cows

The average activity of cows in reticulorumen pH Class 1 was 3.5% lower compared to that of Class 4 and 14.3–14.96% lower compared to that of Classes 1 and 3. In cows from Class 3, we determined the highest temperature of the reticulorumen, and in Class 4, the lowest temperature was found (0.07 °C lower). The differences in arithmetic means were not statistically significant (Table 2).

In Class 2, we found the lowest level of milk (EC) (68.5–70.5), and in the other classes, these were statistically significantly higher (from 70.5 to 72 mS/cm, *p* < 0.05) (Figure 3).

Reticulorumen Class 2 had a lower (*p* < 0.05) RT (3.12% lower compared to Class 3, 12.99% lower compared to Class 4, and 15% lower compared to Class 1). The study showed that the highest levels of milk fat and milk protein and the optimal F/P were in the second class. In Class 1, we found the lowest ratio of milk fat to protein and the lowest concentration of milk lactose. We determined the lowest SCC in the milk of Class 4 and Class 2, while the highest was in Class 3 and Class 1 (Table 2). On the other hand, classes of cows with the highest milk SCC showed the highest electrical conductivity in milk at the udder quarter level (Figure 3).

### 3.3. Correlations of Reticulorumen pH with Indicators from Automatic Milking System (AMS)

Correlation coefficients between reticulorumen pH and indicators from AMS are presented in Figure 4A,B).

Reticulorumen temperature and RT were weakly negatively related with reticulorumen pH (r = −0.131–0.234) and weakly positively correlated with BW and activity of cows (r = −0.051–0.104). MY (r = 0.583, *p* < 0.001), milk lactose (r = 0.240, *p* < 0.05), and F/P (r = 0.250, *p* < 0.05) were positively related with reticulorumen pH and were negatively related with milk protein (−0.304, *p* < 0.01), SCC (−0.329, *p* < 0.05), EC (−0.213–0.498, *p* < 0.05–0.01), and milk fat (−0.042).

The highest blood pH level was determined in reticulorumen classes 2 and 4, and it was lowest in Class 1 (*p* < 0.05). On the contrary, in Class 1 we estimated the highest pCO2 and lowest pO2 and Ca levels. In Class 4, we found the lowest cHCO3-, BE (ecf), TCO2, and Na and the highest levels of K and HCT (Table 3).

Reticulorumen pH was statistically reliable and positively correlated with blood K (*p* < 0.01) and Hct (*p* < 0.001), while it was negatively correlated with pCO2 and TCO2 (*p* < 0.01) as well as with pO2, cHCO3-, BE (ecf), and Na (*p* < 0.05). Data are presented in Figure 5.

## 4. Discussion

### 4.1. Reticulorumen pH as an Indicator of Cow Reproduction Success

The current study indicated that pregnant cows tend to have higher reticulorumen pH during insemination than that of non-pregnant cows. The study findings also indicate that dairy cows with a disturbed rumen metabolism have a low chance of conceiving. Therefore, this highlights that reticuloruminal pH can be used effectively as a predictor for dairy cow reproductive health. According to Inchaisri et al. [17], pH significantly influences conception during insemination. Arguably, a low pH in the reticulorumen increases the temperature of the reticulorumen and abomasum. From this study, the average temperature of the reticulorumen during post insemination until day 170 was considerably higher than that in non-pregnant cows [10]. It was observed that vaginal temperature before estrus was considerably higher than that during the post-ovulation period [18]. During estrus, the average temperature in the reticulorumen increases.

### 4.2. Reticulorumen pH and Health Status of Cows

The available literature indicates that the assessment of ruminal pH is an optimum measure to evaluate the risk of SARA because of variation in dairy cows’ rumen pH [19]. The study findings indicate that dairy cows react uniquely to low pH values of the rumen. Therefore, each cow has varying susceptibility to SARA [20]. Rumination activity and fermentation processes are interconnected. Thus, reduced rumination activity causes lower production of saliva buffering, thereby increasing risk for SARA [21]. The increased rumination activity observed after the calving period is due to the high feed intake during the post-pregnancy process. The accelerated passage rate causes a reduced rumination activity of DMI. Contrary to Pahl et al.’s findings, it was observed that treatment did not affect the rumination patterns of the dairy cows [22]. It was observed that the chew per minute and bolus rumination of dairy cows reduced considerably during the last days before calving and the last days after calving. Similar observations were reported by Schmitz et al. [21].

The study findings indicate that cows with a lower RRpH had a low milk fat/protein ratio, a low lactose concentration, and a high SCC. They also had a low blood pH. Available literature indicates that low ruminal pH triggers the lysing and death of gram-negative bacteria found in the rumen. This action causes an increase in the concentration of lipopolysaccharides, which in turn triggers an increase in the concentration of systemic inflammatory markers, such as cytokines, haptoglobin, and acute protein serum Amyloid A [23]. It is well known that the reticulum has a higher pH level than that of the rumen. Therefore, SARA detection thresholds should be designed in a manner that identifies the localized pH of the reticulum [24]. The current standards for SARA detection involve the use of high-resolution kinetics of rumen pH sensors. However, it was observed that the addition of buffering agents to a high-concentrate diet was effective in preventing milk fat concentration. [25]; this is because it re-established an optimum pH level in the rumen and reticulum.

Feed composition determines the milk fat ratio [26]. The dairy cows under investigation had a low milk fat/protein concentration on most of the test days, which indicated that the energy level of the number of feeds obtained was generally low [27]. This is one of the signs observed in cows presenting with sub-acute rumen acidosis [28]. Dairy cows that have been diagnosed with SARA and non-acute ruminal acidosis generally tend to have lower milk-fat percentages [29]. However, because the disease has different actions on milk fat content per cow, the findings of low milk fat contents concerning feeding composition in most bulk tank testing scenarios remain unclear [30]. The pH of the ruminal fluid was found to be low. This is because the microbes in the rumen break down carbohydrates into short-chain fatty acids at a faster rate than the rumen absorptive rate, outflow, and buffering activity [20]. The reduction of microbial populations in the rumen causes reduced fiber digestion [31]. Consequently, the feed intake reduces [32], further causing a reduction in milk fat production [33]. Altered unsaturated fat bio-hydrogenation processes in the rumen, liver abscesses, systemic and localized tissue inflammation in the rumen papillae, and SARA are the key causes of lameness and horn lesions [34]. Owens et al. [35] argues that chronic and acute acidosis occurs due to the ingestion of diets that contain readily fermented carbohydrates in excess. As a dairy animal adapts to rich concentrates of feeds in their feeding yards, it causes acute acidosis and becomes chronic as the yard feeding continues. In the acute acidosis phase, ruminal acidity and osmolality lead to elevated acids and glucose accumulation, which in turn causes increased damage in the rumen and intestinal wall due to high blood pH and dehydration. These events, if not well managed, can be fatal.

According to the study findings, an increase in RRpH causes an increase in Hct and blood K, and a decrease in BE (ecf), Na, and CO2. According to Giensella et al. [36], it is vital to perform blood gas analyses, as it is a valuable tool, especially during the diagnosis of acidosis. The analysis provides great insight into the extent of acidosis using a noninvasive approach. According to a study conducted by Gokce et al. [37], animals with additional pathological disorders, such as respiratory diseases like pneumonia, tend to display an altered acidotic response. In this study, it was noted that PCO2 differed significantly during the different stages of SARA, which suggested an indication of acute respiratory acidosis. PO2 was observed to decrease statistically during SARA, and it is argued that this pathology is likely due to increased consumption of vascular O2. In this case, decreased PO2 values are associated with increased anaerobic metabolism and O2 consumption [37]. Metabolic disturbances initially present in a hidden form, and their information is associated with problems of fermentation processes in the rumen. It is evident that nutrient conversion is the key precursor of milk production and is largely dependent on rumen fermentation [38]. The functional ability of the mammary gland is directly correlated with the dairy cow’s health status; thus, milk ingredients reflect the level of total metabolism [39]. Therefore, biochemical markers in the milk accurately depict the metabolic status of dairy cows.

## 5. Conclusions

The present study concludes that the interline registered pH of cow reticulum can be used as an indicator of the animal’s health and reproductive status. In pregnant cows, the reticulorumen pH is considerably high during insemination, as compared to that of non-pregnant cows. Cows with a lower RRpH have the lowest milk fat ratio and lactose concentration and the highest SCC. The high RRpH increased the concentration of K and HCT in the blood, but caused a reduction in CO2, BE, and Na. Therefore, reticulorumen pH can be used effectively to predict cow reproductive and health status.

## Figures and Tables

**Figure 1 sensors-20-01022-f001:**
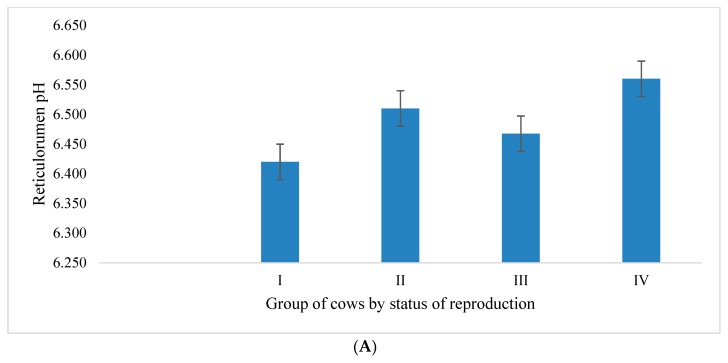
(**A**). Analysis of reticuloromenreticulorumen pH in cows by reproduction status. Group I: 15–30 days postpartum, Group II: 1–34 days after insemination, Group III: 35 days after insemination (non-pregnant), Group IV: 35 days after insemination (pregnant). (**B**). Analysis of reticulorumen pH in cows by status of reproduction. Class 1: pH < 6.22, Class 2: pH 6.22–6.42, Class 3: pH 6.42–6.62, and Class 4: pH > 6.62.

**Figure 2 sensors-20-01022-f002:**
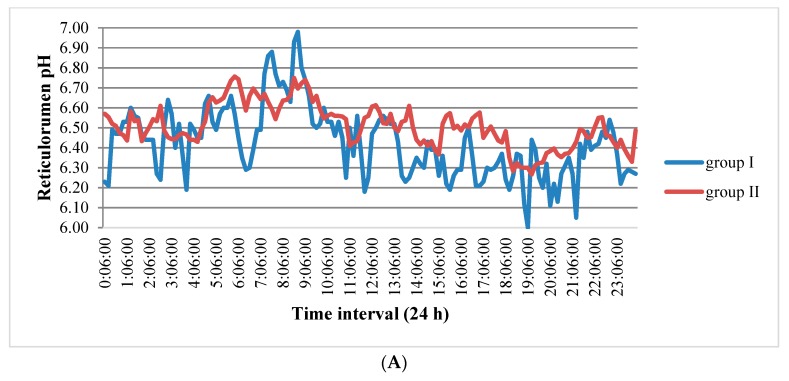
(**A**). Reticulorumen pH changes during 24 h by reproduction status of cows. Group I: 15–30 days postpartum, Group II: 1–34 days after insemination. (**B**). Reticulorumen pH changes over 24 h by reproduction status of cows. Group III: 35 days after insemination (non-pregnant), Group IV: 35 days after insemination (pregnant).

**Figure 3 sensors-20-01022-f003:**
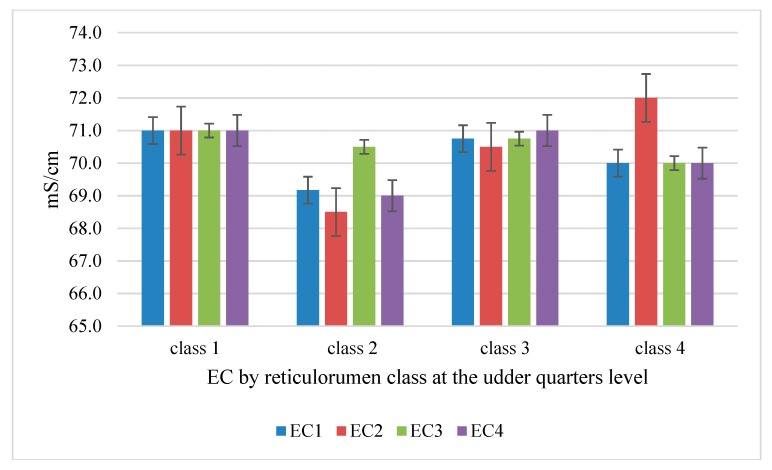
Comparison of electrical conductivity of milk (EC) (mS/cm) by udder quarter level according to reticulorumen pH classes. EC1—front left, EC2—front right, EC3—rear left, EC4—rear right. mS/cm—milisiemens per centimetre.

**Figure 4 sensors-20-01022-f004:**
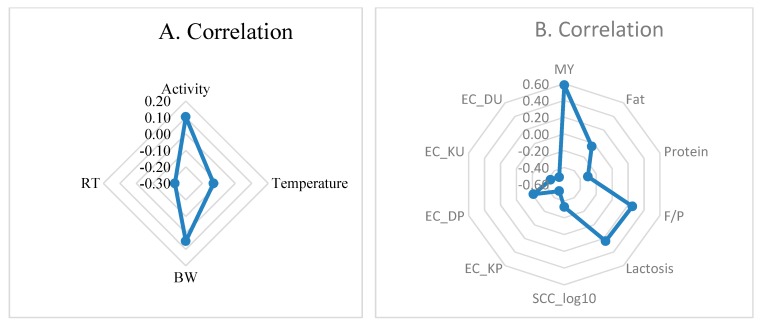
(**A**,**B**). Reticulorumen pH correlations with indicators from AMS. RT—rumination time; BW—body weight; SCC—somatic cell count; EC—electrical conductivity of milk at the udder quarter level (DU—rear right, KU—rear left, DP—front right, KP—front left).

**Figure 5 sensors-20-01022-f005:**
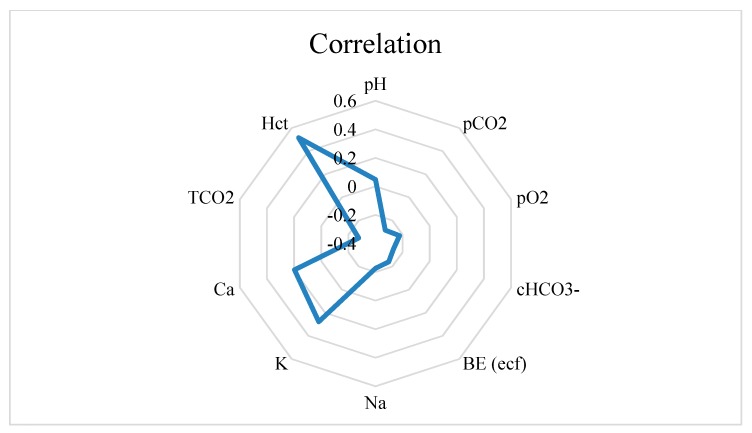
Reticulorumen pH correlations with blood indicators. BE—base excess in blood; PCO2—partial carbon dioxide pressure; PO2—partial oxygen pressure; cHCO3—bicarbonate; pH—hydrogen potential; TCO2—total carbon dioxide carbon; BE (ecf)—base excess in extracellular fluid; Na—sodium; Ca—Calcium; K—potassium.

**Table 1 sensors-20-01022-t001:** Creation of experimental groups.

Group	Days/Status of Reproduction	n	%
I	15–30 d. postpartum	35	36.8
II	1–34 d. after insemination	20	21.1
III	35 d. after insemination (non-pregnant)	20	21.1
IV	35 d. after insemination (pregnant)	20	21.1
Total		95	100.0

**Table 2 sensors-20-01022-t002:** Influence of reticulorumen pH and reproductive status on automatic milking system (AMS) indicators and milk traits of cows.

Reticulorumen pH Class	AMS Parameters (M, SE)			AME Parameters (M, SE)		
1	Activity (steps/hour)	10.24	1.239	Fat (%)	3.58	0.187
2		10.30	0.506		4.58	0.076
3		8.96	0.620		3.93	0.094
4		9.27	0.876		3.93	0.132
1	Reticulorumen temperature (°C)	38.78	0.078	Protein (%)	3.37	0.057
2		38.76	0.032		3.58	0.023
3		38.79	0.039		3.43	0.028
4		38.72	0.055		3.37	0.040
1	BW (kg)	756.00	61.710	F/P	1.06	0.048
2		593.67	25.193		1.28	0.020
3		630.75	30.855		1.15	0.024
4		630.00	43.636		1.17	0.031
1	RT (min/d)	487.00	24.947	Lactose (%)	4.53	0.028
2		423.50	10.185		4.61	0.011
3		436.75	12.474		4.59	0.014
4		478.50	17.640		4.56	0.020
1	MY (kg/d)	37.50	2.214	SCC (tousd/mL)	124.00	222.028
2		41.07	1.067		105.83	90.643
3		37.13	1.307		135.25	111.014
4		49.85	1.849		95.00	156.998

Means with different superscripts among classes are significantly different (*p* < 0.05). M—mean; SE—standard of error of the mean; RT—rumination time; BW—body weight; SCC—somatic cell count; MY—milk yield; F/P—milk fat-protein ratio.

**Table 3 sensors-20-01022-t003:** Influence of reticulorumen pH level on blood indicators in cows.

Reticulorumen pH Class	Blood Parameters (M, SE)			Blood Parameters (M, SE)		
1	pH	7.38 ^a^	0.016	Na	137.00 ^a^	0.601
2		7.43 ^b^	0.005		137.13 ^ab^	0.212
3		7.42 ^b^	0.008		137.25 ^ab^	0.3
4		7.43 ^b^	0.011		136.00 ^ac^	0.425
1	pCO2	49.20 ^a^	2.204	K	3.90 ^a^	0.11
2		45.20 ^b^	0.779		4.10 ^a^	0.039
3		45.13 ^b^	1.102		4.00 ^a^	0.055
4		40.55 ^a^	1.558		4.30 ^b^	0.078
1	pO2	49.90 ^a^	19.062	Ca	1.24 ^a^	0.016
2		67.11 ^a^	6.740		1.13 ^b^	0.006
3		61.45 ^a^	9.531		1.14 ^b^	0.008
4		52.00 ^a^	13.479		1.22 ^a^	0.011
1	cHCO3-	29.30 ^a^	1.288	TCO2	29.20 ^a^	1.257
2		30.23 ^ab^	0.455		29.90 ^ab^	0.445
3		29.03 ^a^	0.644		28.78 ^a^	0.629
4		27.00 ^ac^	0.91		26.75 ^ac^	0.889
1	BE (ecf)	4.20 ^a^	1.372	Hct	24.00 ^a^	0.884
2		5.98 ^ab^	0.485		23.75 ^a^	0.313
3		4.48 ^a^	0.686		26.00 ^b^	0.442
4		2.70 ^ac^	0.97		27.00 ^b^	0.625

a,b,c Means with different superscripts among classes are significantly different (*p* < 0.05). M—mean; SE—standard of error of the mean; BE—base excess in blood; PCO2—partial carbon dioxide pressure; PO2—partial oxygen pressure; cHCO3—bicarbonate; pH—hydrogen potential; TCO2—total carbon dioxide carbon; BE (ecf)—base excess in extracellular fluid; Na—sodium; Ca—Calcium; K—potassium.

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
