# Peer review of "Inline Reticulorumen pH as an Indicator of Cows Reproduction and Health Status"

_sensors, 2020, doi:10.3390/s20041022_

Round 1

Reviewer 1 Report

Major remark:

I believe the science presented here, may not be a scope for this journal. The results and discussion presented are for more targeted audience of dairy, animal science and veterinary medicine. The editor can take the final decision.

Minor remarks

·         Manuscripts lacks consistency, because of this its hard to understand your science for this journal audience.

·         Include list of acronyms, readers will find it helpful

·         No consistency - some with “Ph” and others with “pH”, check all the typos across the manuscript

·         Line 54: incomplete “can be used to assess the disease, so future studies comparing the data findings using many animals”

·         For all figures - Check font style, font sizes, for graphs add solid lines for x and y axis, include figure legend (short description of the figure) not just the figure title. Mention what type of figure chart it is, e.g. Fig 3 and 4

·         Fig 2: what is x and y axis?

·         Describe Tables and figures

o   What is M, SE, BE (ecf) don’t assume the reader can figure it out? You have submitted this article to an inter disciplinary journal, some of the readers don’t have the background knowledge about dairy production. What are superscripts a, b and c, clearly describe the differences

·         Line 179, 246 : What is “EC” “RR”? you mentioned RRpH but not RR,

·         Line 225: “dairy’s reproductive health”?  which dairy’s?

Author Response

Dear Reviewer,

Thank You for Your a very valuable review. Corrections and comments are presented in the table. All correction in manuscript highlighted in green. All language editing correction are presenting in track changes.

Best Regards,

Prof. Ramunas Antanaitis

Question

Answer

I believe the science presented here, may not be a scope for this journal. The results and discussion presented are for more targeted audience of dairy, animal science and veterinary medicine. The editor can take the final decision.

This manuscript are preparing for special Issue entitled "Water and Health pH Sensors", to be published in the journal Sensors. Sensors for measuring pH have wide applications, ranging from water to health to environmental monitoring. pH is an important indicator of the quality of water and health conditions. Recently, considerable interest has been shown for wearable sensors for the early prediction of diseases through the non-invasive analysis of bodily fluids like reticulorumen, blood and ect.

Manuscripts lacks consistency, because of this its hard to understand your science for this journal audience.

This manuscript has undergone English language editing by MDPI. The text has been checked for correct use of grammar and common technical terms, and edited to a level suitable for reporting research in a scholarly journal. MDPI uses experienced, native English speaking editors. Full details of the editing service can be found at https://www.mdpi.com/authors/english.

All correction are presenting in track changes

Include list of acronyms, readers will find it helpful

Included – L314 – 355

No consistency - some with “Ph” and others with “pH”, check all the typos across the manuscript

Corrected

  Line 54: incomplete “can be used to assess the disease, so future studies comparing the data findings using many animals”

Corrected to -  “However, there is limited information on how the two parameters can be used to assess the disease, so future studies comparing the data findings using many animals are needed”

  For all figures - Check font style, font sizes, for graphs add solid lines for x and y axis, include figure legend (short description of the figure) not just the figure title. Mention what type of figure chart it is, e.g. Fig 3 and 4

Corrected

Fig 2: what is x and y axis?

Corrected

Describe Tables and figures

What is M, SE, BE (ecf) don’t assume the reader can figure it out? You have submitted this article to an inter disciplinary journal, some of the readers don’t have the background knowledge about dairy production.

Corrected

What are superscripts a, b and c, clearly describe the differences

Corrected. Means with different superscripts among classes are significantly different (P <0.05)

Line 179, 246 : What is “EC” “RR”? you mentioned RRpH but not RR,

L 179 - corrected to “electrical conductivity of milk (EC)”

L246 – corrected to RRpH

Line 225: “dairy’s reproductive health”?  which dairy’s?

corrected to “dairy cows reproductive health”

Reviewer 2 Report

The paper describes the use of pH as a biomarker for the monitoring of cattle health.  The paper would be of interest to the readership of sensors.  However, before I can recommend publication, the following points must be addressed:

Can you refer to a pH reading as a biomarker? Biomarkers refer to biomolecules that are either present or change in concentration with biological fluids.

I’m detecting a few grammatical errors.  I recommend that the authors have their manuscript proofread by a native speaker.

Line 36  “and use of online-time milk analyzer”.

Line 41: (pp.403-425)  Has this manuscript been plagiarized from somewhere else?   

The time period is irrelevant to the study. I recommend that the authors remove it.

Line 101: The authors should give more information on “SmaX-tec boluses” Who makes this device and where it is manufactured?

Line 103:  “Using the instruction manual, boluses were put into the cow’s reticulorumens”. The data was collected using specific antennas of the SmaX-tec device.  How long was it placed their for?

Line 108: “More so, it is shock-proof to rumen fluid”.  You mean it is resistant to rumen fluid

Fig 1  B  How can Class III have 100% with a pH of <6.22 and have about 50% with a pH of between 6.42-6.62 and 50% with a pH >6.62.  I think there is something wrong with this graph or it is not clear.

Fig2:  x axis units should be put as 1, 2 3 hour etc.  It would be clearer for the reader. Also the graph axis needs labelling with units

The authors mention that one group fluctuates more than the others.  I cant quite see it from this graph.  I would recommend that the authors calculate the mean average and standard deviation to show the amount of fluctuation.

Line 167: Which is which.  You need to state which group refers to which cows throughout the manuscript for clarity to the reader.

Figure 3 EC units in y axis?

Author Response

Dear Reviewer,

Thank You for Your a very valuable review. Corrections and comments are presented in the table. All correction in manuscript highlighted in green. All language editing correction are presenting in track changes.

Best Regards,

Prof. Ramunas Antanaitis

Question

Answer

Can you refer to a pH reading as a biomarker? Biomarkers refer to biomolecules that are either present or change in concentration with biological fluids.

Authors totally agree with You, and corrected to “indicator” in all text.

I’m detecting a few grammatical errors.  I recommend that the authors have their manuscript proofread by a native speaker.

This manuscript has undergone English language editing by MDPI. The text has been checked for correct use of grammar and common technical terms, and edited to a level suitable for reporting research in a scholarly journal. MDPI uses experienced, native English speaking editors. Full details of the editing service can be found at https://www.mdpi.com/authors/english.

All correction are presenting in track changes

Line 36 “and use of online-time milk analyzer”.

Corrected

Line 41: (pp.403-425)  Has this manuscript been plagiarized from somewhere else?  

This statement is made shorter – “ The sensors in these applications provide useful data that can be used by farmers to identify livestock that needs special care before the health conditions worsen. [5] (pp. 403-–404)”

The time period is irrelevant to the study. I recommend that the authors remove it.

Removed

Line 101: The authors should give more information on “SmaX-tec boluses” Who makes this device and where it is manufactured?

Corrected to “ smaXtec animal care GmbH, Austria”

Line 103:  “Using the instruction manual, boluses were put into the cow’s reticulorumens”. The data was collected using specific antennas of the SmaX-tec device.  How long was it placed their for?

Boluses were into cows reticulorumens from 20190201 till 20190901 and date were  registered during the investigation day, except for pregnant and non-pregnant cows, whose data was were registered during the insemination day.

Line 108: “More so, it is shock-proof to rumen fluid”.  You mean it is resistant to rumen fluid

Yes – it is resistant to rumen fluid. Corrected to “ ...it is was is resistant to rumen fluid”

Fig 1  B  How can Class III have 100% with a pH of <6.22 and have about 50% with a pH of between 6.42-6.62 and 50% with a pH >6.62.  I think there is something wrong with this graph or it is not clear.

Corrected.    We found (Fig 1B) that all pregnant cows (group IV, n=20) belonged to the third class according to their reticulorumen pH, which ranged from 6.42 to 6.62 (50.00% of the animals in this class were group III cows).

Fig2:  x axis units should be put as 1, 2 3 hour etc.  It would be clearer for the reader. Also the graph axis needs labelling with units

Corrected and  divided in to two Fg – Fg2A and Fig2B.

The authors mention that one group fluctuates more than the others.  I cant quite see it from this graph.  I would recommend that the authors calculate the mean average and standard deviation to show the amount of fluctuation.  

The data in Fig. 2A showed that the pH of the first group (15-30 days postpartum) changed from 6 to 6.98 during the day. The range of changes in this indicator was 2 - 2.24 times higher compared to cows of other groups.

Line 167: Which is which.  You need to state which group refers to which cows throughout the manuscript for clarity to the reader.

Corrected

Figure 3 EC units in y axis?

Corrected

Reviewer 3 Report

The experimental design of this study is questionable. There is no novelty and the materials and methods and especially results, are very poor. The presentation of results is unclear and their discussion lacks adequate scientific quality. The statistics need serious revision and ROC curves and cross validation are needed. Unfortunately, I can't suggest this paper for publication in this present form.

Author Response

Dear Reviewer,

Thank You for Your a very valuable review. Corrections and comments are presented in the table. All correction in manuscript highlighted in green. All language editing correction are presenting in track changes.

Best Regards,

Prof. Ramunas Antanaitis

Question

Answer

The experimental design of this study is questionable.

This manuscript has undergone English language editing by MDPI. The text has been checked for correct use of grammar and common technical terms, and edited to a level suitable for reporting research in a scholarly journal. MDPI uses experienced, native English speaking editors. Full details of the editing service can be found at https://www.mdpi.com/authors/english.

All correction are presenting in track changes

This manuscript are preparing for special Issue entitled "Water and Health pH Sensors", to be published in the journal Sensors. Sensors for measuring pH have wide applications, ranging from water to health to environmental monitoring. pH is an important indicator of the quality of water and health conditions. Recently, considerable interest has been shown for wearable sensors for the early prediction of diseases through the non-invasive analysis of bodily fluids like reticulorumen, blood and ect.

There is no novelty and the materials and methods and especially results, are very poor. The presentation of results is unclear and their discussion lacks adequate scientific quality.

Materials and methods are corrected and correction are presented in green and track changes.

The statistics need serious revision and ROC curves and cross validation are needed.

Data interpretation is partially adjusted. It is in manuscript highlighted in green

Round 2

Reviewer 2 Report

The authors have addressed all the points apart from the comment on the use of (pp4759-4773).  I assume that the authors are referring to the page numbers of their citations rather than any case of plagiarism.  Can the authors remove these page numbers from their manuscript as this is not usual convention to list the page numbers of the citations in the text of the manuscript.  

Author Response

Dear Reviewer,

Thank You for review. The page number from our manuscript removed.

Best Regards,

Prof. Ramunas Antanaitis